# Pangenomic Characterization of *Campylobacter* Plasmids for Enhanced Molecular Typing, Risk Assessment and Source Attribution

**DOI:** 10.3390/pathogens14090936

**Published:** 2025-09-16

**Authors:** Lucas Harrison, Sampa Mukherjee, Cong Li, Shenia Young, Qijing Zhang, Shaohua Zhao

**Affiliations:** 1U.S. Food and Drug Administration, Center for Veterinary Medicine, Laurel, MD 20708, USA; sampa.mukherjee@fda.hhs.gov (S.M.); cong.li@fda.hhs.gov (C.L.); shenia.young@fda.hhs.gov (S.Y.); shaohua.zhao@fda.hhs.gov (S.Z.); 2College of Veterinary Medicine, Iowa State University, Ames, IA 50011, USA; zhang123@iastate.edu

**Keywords:** plasmid, *Campylobacter*, pangenome, molecular typing, AMR, pMLST

## Abstract

Plasmid-mediated dissemination of antimicrobial resistance (AMR) and virulence genes plays a critical role in enhancing the adaptive potential of *Campylobacter* spp. While *Campylobacter* plasmids of concern are commonly classified as pTet, pVir, pCC42 or a large plasmid encoding a T6SS (pT6SS), existing classification systems often lack the resolution to capture intra-group diversity. Here we demonstrate a plasmid typing approach with enhanced discriminatory power that categorizes these major plasmid groups into discrete subgroups and strengthens risk-assessment investigations. Pangenomic analysis of 424 *Campylobacter* plasmid sequences revealed 30 distinct plasmid groups. The four major groups above accounted for 74.3% of the dataset. Within these major groups, 177 plasmid type-specific loci were used to define 16 subgroups. pTet plasmids were subdivided into 5 subgroups, with subgroup 3 enriched in *C. coli*. pVir plasmids formed 3 subgroups, with only subgroup 3 harboring the *tet(O)* genes. The 5 pCC42 subgroups displayed *Campylobacter* species specificity while the 3 pT6SS subgroups encoded distinct AMR profiles. This high-resolution typing approach provides a unified and scalable method to characterize *Campylobacter* plasmid diversity and identifies genetic markers critical for pathogen surveillance, source attribution and mitigation strategies employed to safeguard human and animal health.

## 1. Introduction

*Campylobacter* bacteria are human pathogens and part of the community of commensal microorganisms in food animals, wild animals and companion animals [1,2]. *Campylobacter* species have been isolated from a wide range of sources including live animals, retail meats, freshwater reservoirs, topsoil and even vegetation [2,3,4]. While many *Campylobacter* strains reside asymptomatically in immunocompetent veterinary hosts, some strains exhibit increased pathogenicity [5,6]. Although sporadic in nature, campylobacteriosis remains the most common form of human bacterial foodborne enteritis [5]. The rarity of outbreak events indicates that the progression to disease is not always dependent on highly clonal populations of pathogenic strains that are adapted to survive in a broad range of environments. Rather, it indicates that campylobacteriosis is dependent on the broad genetic diversity within the *Campylobacter* population that allows individual strains to capitalize on the opportunities presented in their unique microenvironment. These sporadic infections, caused by genetically diverse strains, also challenge efforts to identify the specific genetic elements that differentiate highly pathogenic strains from those with decreased pathogenicity [7].

One approach to differentiate *Campylobacter* strains involves the use of whole-genome sequencing (WGS) to identify the core set of genes conserved among the two most clinically relevant species *C. coli* and *C. jejuni* [8]. This effort led to the development of a 1343 loci core genome multilocus sequence typing (cgMLST) schema [8]. The cgMLST typing scheme has been instrumental in associating genomic profiles with AMR gene carriage, geographical location, isolation host and pathogenic risk [9,10]. However, the core loci shared among non-*jejuni* or -*coli Campylobacter* species may be as low as 500 loci [11], limiting the typing method’s utility for characterizing clinically relevant strains from other species. Furthermore, cgMLST does not fully account for the diversity of virulence and AMR genes located in the accessory genome or on mobile genetic elements [12].

The *Campylobacter* cgMLST loci contain alleles commonly associated with virulence or antimicrobial resistance, including but not limited to *cdtB*/CAMP0068 [13], *cmeC*/CAMP0330 [13], *cmeB*/CAMP0331 [13], *cmeA*/CAMP0332 [13], PEB4/CAMP0554 [14], *gyrA*/CAMP0950 [15], *cheY*/CAMP1039 [16,17], *htrB*/CAMP1055 [18], CJ1135/CAMP1056 [19], *porA*/CAMP1178 [20] and *kpsF*/CAMP1354 [21]. While other mechanisms of resistance and virulence are encoded among the chromosomes of other *Campylobacter* species, these elements are found outside the core genome [22,23,24]. In some cases, these accessory elements reside on mobile elements such as plasmids, and can encode genes associated with resistance to tetracyclines, macrolides, aminoglycosides, chloramphenicol, cephalosporins and other β-lactam antibiotics [9]. Further, plasmid-borne genes can enhance virulence by providing *Campylobacter* strains with type IV (T4SS), type VI (T6SS) and other secretion systems [25]. These systems mediate host cell adhesion, injection of protein effector molecules into host cells and horizontal DNA transfer between bacteria [26]. Plasmid-mediated transfer of DNA through secretion systems can augment the AMR profile and virulence potential of strains beyond what is accounted for in the core genome [27,28].

Despite their potential to enable antimicrobial resistance or increase the pathogenicity of *Campylobacter* strains, *Campylobacter* plasmids are not as well characterized as plasmids from Gammaproteobacteria. Three of the major *Campylobacter* plasmid groups have been named: pTet, pVir and pCC42. The fourth major *Campylobacter* plasmid group is often referred to as the *Campylobacter* megaplasmid and encodes a T6SS [29]. pTet and pVir are the two most commonly reported *Campylobacter* plasmids [30]. pTet plasmids are carried by both *C. jejuni* and *C. coli*, maintain a size of ~40–55 kb, and commonly encode the *tet(O)* tetracycline resistance gene [31]. pVir plasmids carry a T4SS, are found in both *C. jejuni* and *C. coli* and contribute to the virulence factors that allow *Campylobacter* bacteria to survive in the host [30]. The first pCC42 plasmid was recovered from a *C. coli* isolate from a 42-year-old patient, and while pCC42 plasmids are most prevalent among *C. coli*, they have been identified in *C. jejuni* as well [31]. The pCC42 plasmids are smaller (<30 kb) and also encode a T4SS, but generally lack AMR genes [32]. A newer group of large (190 kb) T6SS-encoding plasmids (pT6SS) has been linked to enhanced red blood cell-specific cytotoxicity [29]. Although each group is defined by a core genetic background, individual plasmids carry unique gene profiles that highlight the sequence diversity found within these plasmid groups [25].

These four plasmid groups harbor functionally important genes, yet there is no unified typing scheme to define their core genetic elements or resolve their internal diversity. While the *Campylobacter* chromosome can be characterized with cgMLST, there is no analogue for *Campylobacter* plasmids that aids in plasmid tracking, risk assessment or molecular epidemiological studies. Here we use a plasmid characterization tool, Lociq [33], to perform a pangenomic analysis of a large dataset of *Campylobacter* plasmids from human, animal and environmental origin. Our goals are to (1) define the core loci for each of the major plasmid types, (2) develop a high-resolution typing framework to distinguish plasmid subgroups, and (3) identify signature genetic elements to facilitate risk assessment and source attribution of plasmid-associated AMR and virulence factors. This approach provides a scalable and informative method to investigate *Campylobacter* plasmid diversity and its implications for human and animal health.

## 2. Materials and Methods

### 2.1. Campylobacter Chromosome and Plasmid Sequencing and Assembly

Plasmid sequences were recovered from two sources. The dataset of 83 *Campylobacter* plasmids from the US National Antimicrobial Resistance Monitoring System (NARMS) used in this study was represented by a combination of plasmids that were sequenced onsite for this study and closed *Campylobacter* plasmid sequences from the NARMS program that had been previously uploaded to the National Center for Biotechnology Information (NCBI) [34]. The dataset of 341 *Campylobacter* plasmids from PLSDB were recovered by downloading a local copy of the PLSDB v2023_11_03_v2 [35] database and extracting all sequences from entries associated with a *Campylobacter* isolation source. DNA from 58 NARMS [36] *Campylobacter* isolates was recovered using the Wizard^®^ Genomic DNA Purification Kit (Promega, Madison, WI, USA). WGS was performed on an Oxford Nanopre Technology (ONT) MinIon, version MIN-101B, using the Flow Cell R9.4.1 and rapid barcoding kit, SQK-RBK004 (Oxford Nanopore, Oxford, UK), following manufacturer’s instructions. Sequences were assembled with Trycycler v0.5.5 [37] using FLYE v2.9.3 [38], Raven v1.8.3 [39] and miniasm v0.3-r179 [40] as the primary sequence assemblers and specifying an approximate genome size of 1.8 Mb. Assemblies were first polished with Medaka v1.11.3 [41] and Polypolish v0.6.0 [42] when short-read sequences were available through NCBI, and then screened for regions of low coverage using samtools v1.18 [43] and bedtools v2.27.1 [44]. Contigs larger than 1.4 Mb were identified as chromosomal sequences, while smaller contigs with lower depth of coverage than the chromosomal contigs were removed. The remaining contigs were processed as plasmid contigs.

### 2.2. Plasmid Feature Annotation

Plasmid assemblies were annotated with PROKKA v1.14.6 set to include non-coding RNAs [45] and the resulting ordered protein sequence files were analyzed using MacSyFinder v2.1.4 with the TXSScan v1.1.3 and CONJScan v2.0.1 libraries [46,47,48] to identify molecular secretion and conjugation systems. Additional plasmid-borne virulence and antibiotic resistance genes were identified through AMRFinder Plus [49] v3.10.45 with the -plus parameter included. Plasmid sequences were characterized using the MOB-suite mob_typer 3.1.9 [50,51].

### 2.3. Plasmid Group Characterization

Within this study, *Campylobacter* plasmid type refers to the accepted plasmid classification system that contains but is not limited to pTet, pVir and pCC42. *Campylobacter* plasmid group, however, refers to the plasmid groupings derived from our dataset during this study using the following process. *Campylobacter* plasmids were analyzed and organized into groups by gene content using the Lociq [33] plasmid analysis method. This process included the generation of the pangenome of combined coding and intergenic sequences with the Roary v3.13.0 [52] and Piggy v1.5 [53] programs, respectively. The pangenome was translated into a presence/absence matrix of genetic elements and the pairwise Jaccard distances were calculated between all plasmids in R [54]. We then applied the elbow method to the sum of squared Jaccard distance values to determine the number of plasmid groups contained in the dataset [55]. Images for the elbow plots were generated in the core R environment. Plasmid typing loci were selected using the Lociq default values of >90% prevalence within the plasmid group and <10% outside the plasmid group.

*Campylobacter* plasmid groups were subdivided by applying the elbow method described above to the set of plasmids within each group. The plasmid typing loci for the plasmid subgroups were determined in a similar two-step process. First, a candidate library of loci was generated by identifying the loci with a <10% prevalence outside the plasmid group to filter out loci that were likely to be more prevalent in other plasmid groups. Second, the prevalence of the loci from this subset was evaluated for each plasmid subgroup. Loci with a prevalence >90% in any of the plasmid subgroups were identified as being a plasmid typing locus for that plasmid subgroup. Plasmid subgroup typing results were compared to mob_typer assignments with the adjusted Rand index in R using the mclust v6.1.1 package [56].

Plasmid typing loci were checked for uniqueness by generating a pairwise percent identity matrix of their primary protein sequences using the R package bio3d v2.4-4 [57]. Loci with a greater than 70% identity of their aligned regions were removed from the set of plasmid typing loci. Functional annotation of the remaining coding loci was performed by querying the protein sequences against the Uniprot 2024_04 database and with BlastKOALA v3.0 through their respective web interfaces [58,59]. Consensus sequences of the intergenic loci were generated through EMBOSS v6.6.0.0 *cons* program using default parameters [60].

## 3. Results

Eighty-three closed plasmid sequences were recovered from 58 *Campylobacter* isolates collected through the NARMS program. Pangenomic analyses (Figure 1) revealed that the *Campylobacter* plasmids were organized in 8 groups (Appendix A). The largest 4 groups accounted for 72 of the 83 plasmids (86.7%) and contained between 16 and 21 plasmids each. Plasmids from each of these major plasmid groups aligned with one of four known *Campylobacter* plasmid types: pTet, pVir, pCC42 and pT6SS. As our initial analysis identified 8 plasmid groups, the remaining 4 plasmid groups contained the 11 plasmids left in the dataset.

Next, we evaluated if the pattern of 4 major *Campylobacter* plasmid groups would emerge when the pangenome was reconstructed using a combined dataset of 83 NARMS plasmid sequences and 341 plasmid sequences from an external database representing 15 *Campylobacter* species. The combined dataset of 424 *Campylobacter* plasmids was divided into 30 plasmid groups, and 315 of the 424 plasmids were sorted into the 4 major *Campylobacter* plasmid groups representing pTet, pVir, pCC42 and pT6SS (Appendix A). Alignment of the plasmid sequences to representative plasmids from the four major groups showed that plasmids in the pTet, pVir and pCC42 groups aligned to the reference with >89% identity (Table 1). The pT6SS plasmids only aligned to the reference with 65% identity. However, 36.1% of the pTet reference plasmid aligned to the pT6SS plasmid sequences, indicating the presence of conserved regions shared between the two plasmid groups.

All plasmids from the 4 major *Campylobacter* plasmid groups that were represented in the NARMS dataset were mapped to their corresponding group in the combined dataset (Figure 2). A visual inspection of the distribution of loci revealed that each major plasmid group was made up of several, smaller subgroups. This pattern of subdivision was most notable in the pCC42 plasmids of the NARMS dataset where half of the pCC42 plasmids contained loci that were absent in the other pCC42 plasmids (Appendix A). Next, we generated elbow plots of the major *Campylobacter* plasmid groups to determine whether the groups warranted further subdivision, and if so, to identify the number of subgroups present in each (Appendix A). The pTet and pCC42 groups were each divided into 5 subgroups while the pVir and pT6SS groups were each divided into 3 subgroups.

Pangenomic analysis using the Lociq program revealed 177 genetic markers that defined the 16 plasmid subgroups: 5 pTet subgroups, 5 pCC42 subgroups, 3 pVir subgroups and 3 pT6SS subgroups (Appendix A). Typing of these subgroups showed strong agreement with the secondary cluster ID assigned by the MOBTyper program (adjusted Rand index, 0.669) (Appendix A). Protein coding regions of plasmid loci were evaluated in a percent identity matrix to identify loci pairs with a sequence similarity greater than 70% shared between plasmid subgroups (Appendix A). Only one typing locus was identified in multiple groups; the *hicB* antitoxin gene was shared between the pTet and the pT6SS plasmids. Of the protein coding regions, 108 were initially annotated as hypothetical proteins. Functional annotation of these 108 loci was performed with BlastKOALA. This approach provided multiple annotations per locus resulting from multiple functional sites on a given locus. The best match for each locus was selected as the annotation (Appendix A).

Group 1 represented the 165 pTet plasmids collected from all isolation sources represented in the dataset (Appendix A). 84.8% of the pTet plasmids encoded a pT4SSt system and 9.8% of the pTet plasmids carried type T T4SS genes. Only 3.0% of pTet plasmids lacked any known bacterial secretion systems. The pTet group accounted for the plasmids with the greatest prevalence and diversity of AMR genes. Specifically, 96.4% of pTet plasmids bore AMR genes and 16/17 AMR genes found in the dataset were identified in the pTet group.

Plasmid-encoded AMR genes were unequally distributed among the 5 pTet subgroups (Figure 3). While the pTet.5 subgroup contained plasmids that only encoded the *tet(O)* AMR gene, at least 7 AMR genes were identified in each of the 4 remaining pTet subgroups. Of these subgroups, pTet.3 showed the greatest diversity of AMR genes with 12 unique AMR alleles present in the subgroup (Figure 3). Additionally, all pTet subgroups from *C. coli* showed a greater prevalence of AMR genes than pTet plasmids recovered from *C. jejuni*. The importance of this is highlighted in pTet.3 which not only contained the highest number of AMR genes, but also contained the greatest fraction of pTet^+^
*C. coli* isolates of human origin (Appendix A). Of note, the large number of AMR genes represented in this subgroup was driven by a set of plasmids recovered from chicken isolates that bore *aad9*, *ant(6′)I-a*, *aph(2”)-Ig*, *aph(3′)-IIIa*, *sat4* and *tet(O)* (Appendix A).

A total of 38 loci were identified across the 5 pTet subgroups (Appendix A); 29 loci encoded proteins while 9 loci were intergenic sequences. Fourteen of the protein coding loci were specific to a single pTet subgroup (Appendix A). pTet.2 plasmids were identified by the presence of genes encoding proteins with amino acid sequence similarity to a demethylase (Uniprot ID: A0A1C7H5E6), an EexN family lipoprotein (Uniprot ID: A0A4U8UF75), a toprim domain-containing protein (Uniprot ID: A0A329ZV89) and a variant of the conjugal transfer protein TraJ (Uniprot ID: A0A329ZX00). pTet.3 plasmids were defined by genes encoding unique variants of TrbL (Uniprot ID: A0A329ZY80) and TraJ (Uniprot ID: A0A329ZX00) as well as a DUF2306 domain-containing protein (Uniprot ID: A0A2U2XA67), a ribonuclease P protein component (Uniprot ID: A0A926MQ05), and a large polyvalent protein-associated domain-containing protein (Uniprot ID: V8CF05). pTet.4 plasmids were identified by genes encoding DNA topoisomerase 3 (Uniprot ID: B9KGJ1), a TraC variant (Uniprot ID: A0A4U7BAF4) and an uncharacterized protein. Only one gene encoding a site-specific recombinase/resolvase (Uniprot ID: B9KGG8) was indicative of pTet.5 plasmids while pTet.1 plasmids contained no loci that were unique to their specific subgroup.

*Campylobacter* plasmid group 2 (pCC42) contained 67 plasmids recovered from all isolation sources represented in this study (Appendix A). The pCC42 plasmids encoded either a T4SS with a functional protein transfer system (65/67) or the related genetic mechanisms for plasmid transfer (2/67) (Appendix A). The pCC42 group was divided into 5 separate subgroups using 36 protein-coding loci and 10 intergenic loci (Appendix A). Sequence analysis revealed seven instances where two or more protein-coding loci from pCC42 subgroup plasmids shared >70% sequence identity (Appendix A). The sequence similarity among loci suggests that the functional roles of these variants are specialized for each subgroup (Table 2). The prevalence of pCC42 by species was 35.8% in *C. jejuni*, 62.7% in *C. coli* and 1.5% in *C. armoricus* (Appendix A). The pCC42 subgroups demonstrated greater species specificity than other *Campylobacter* plasmid groups (Appendix A). Subgroups pCC42.1 and pCC42.4 were represented by 83.3% and 100% *C. jejuni*, respectively. *C. coli* was the dominant species in pCC42.2, pCC42.3 and pCC42.5, representing 90.9%, 100.0% and 71.4% of the subgroups, respectively.

The Group 3 plasmids (pVir) contained 31 plasmids and were recovered from human, chicken and environmental sources (Appendix A). All pVir plasmids encoded either a protein modified T4SS (TXSScan: pT4SSt) or related mechanisms of plasmid transfer. *tet(O)* was the only AMR gene present in the pVir group, and only 3 (9.7%) of the plasmids bore the gene. A 4904 bp sequence surrounding *tet(O)* was conserved in all pVir *tet(O)*^+^ plasmids. The aligned region of this sequence had >99% sequence identity to a 7952 bp region of the reference pTet plasmid (Figure 4). While the 7952 bp aligned region found on the reference pTet plasmid contained both *tet(O)* and *repA*, *repA* was missing from the *tet(O)*^+^ pVir plasmids (Figure 4). Even though this ~5 kb sequence was found in multiple *Campylobacter* plasmid groups, no insertion sequences were detected in the gene-cassette flanking regions or in the *repA* region of the pTet plasmid to explain this pattern of recombination.

Twenty-five coding region loci and 3 intergenic sequence loci were identified as indicative of the pVir plasmids (Appendix A). While the pVir group was divided into 3 subgroups, the subgroup pVir.2 contained only a single plasmid recovered from a *C. novaezeelandiae* isolate. The pVir.1 plasmids contained genes that encoded proteins with sequence similarity to a stage 0 sporulation protein A homolog (Uniprot ID: A0A1D8GI77), a plasmid replication initiation protein (Uniprot ID: A0AA43I9M7) and 4 additional uncharacterized proteins. The pVir.3 plasmids contained a different set of genes that encoded proteins with sequence similarity to a P-type type IV conjugative transfer system coupling protein TraG/VirD4 (Uniprot ID: A0A7H9CK28), a plasmid mobilization relaxosome protein MobC (Uniprot ID: A0A2P8QYF6), a T4SS protein VirB5 (Uniprot ID: A0A3D8IGZ2) and three additional uncharacterized proteins. Further, the pVir.3 subgroup was the only pVir group to contain plasmids encoding the *tet(O)* gene mentioned above. All three pVir.3-*tet(O)*^+^ plasmids were recovered from chicken isolates; two isolates were *C. coli* and one was *C. jejuni*. However, even though the pVir.3 subgroup contained *tet(O)*, no pVir.3 plasmids were recovered from a human-origin *Campylobacter* isolate.

Group 4 (pT6SS) contained the largest plasmids (~123 kb) compared with all *Campylobacter* plasmid groups. These pT6SS plasmids were recovered from *Campylobacter* strains isolated from human sources, avian sources and a single ursine source (Appendix A). These large plasmids had the greatest number of typing loci containing 46 coding loci and 19 intergenic loci (Appendix A). The pT6SS plasmids were divided into three subgroups; pT6SS.1 was defined by 21 coding and 9 intergenic loci, pT6SS.3 was defined by 13 coding and 4 intergenic loci and the pT6SS.2 plasmids lacked any subgroup-specific typing loci.

Despite the diversity of typing loci among pT6SS plasmids, all three subgroups contained hybrid pT6SS/pTet plasmids (Appendix A). In total, 16/52 of the pT6SS plasmids contained at least 70% of the reference pTet plasmid sequence. These 16 plasmids were on average 45 kb larger than the set of pT6SS plasmids that contained less than 70% of the reference pTet plasmid sequence (Figure 5). Additionally, in comparison to a 96.4% carriage rate of *tet(O)* in pTet plasmids, only 87.5% of the hybrid pT6SS/pTet plasmids bore the *tet(O)* gene.

Aside from *tet(O)*, the distribution of AMR genes among the pT6SS plasmids showed plasmid subgroup specificity (Appendix A). *tet(O)* was the only AMR gene present among the pT6SS.1 plasmids while only two plasmids in pT6SS.2 carried AMR genes: NZ_CP082878.1 carried *tet(O)* and pN17C264.1 carried both *tet(O)* and aph(3′)-IIIa. Subgroup 3, however, had the greatest prevalence and diversity of AMR genes among the pT6SS plasmids. Genes encoding Aad9, Aph(2′′)-If, Aph(3′)-IIIa, Aph(3′)-VIIa and Tet(O) were all found in the pT6SS.3 subgroup and seventeen of the 22 pT6SS.3 plasmids carried at least one of these 5 AMR genes. Notably, the aminoglycoside resistance gene *aph(3′)-VIIa* was only found in the pT6SS.3 subgroup among all 424 plasmids in the combined *Campylobacter* plasmid dataset.

A comparison of the four major *Campylobacter* plasmid groups by isolation source revealed that the respective plasmid subgroups were not found equally among either the *Campylobacter* species or the host organisms from which the *Campylobacter* isolates were recovered (Figure 6). For example, 12 of the 15 plasmid subgroups demonstrated *Campylobacter* species specificity; only the pTet.3, pTet.5 and pVir.3 subgroups were comparably represented by both *C. coli* and *C. jejuni*. Further, the *Campylobacter* species composition within the set of isolates recovered from humans differed by plasmid subgroup. Thirteen of the 15 plasmid subgroups contained plasmids recovered from human isolates. In 9 of these 13 subgroups, plasmids from the human isolates were recovered exclusively from either *C. jejuni* or *C. coli*. Only four plasmid subgroups, pCC42.1, pTet.2, pTet.3 and pTet.5, contained plasmids from human isolates of both *C. jejuni* and *C. coli*.

The pTet, pVir, pCC42 and pT6SS plasmid groups mentioned above represent the four major groups out of the 30 plasmid groups that were identified. Groups 5, 9, 10, 12 and 18 were the largest of the remaining groups and contained more than five plasmids each (Appendix A). Plasmid groups 5, 9 and 10 contained smaller plasmids (<5 kb or less) from *C. coli* and *C. jejuni* that were recovered from human, poultry and unknown sources and lacked any known AMR genes (Appendix A). Plasmids from group 5 were identified as mobilizable plasmids, but no mechanisms of transfer or virulence were detected in plasmids from groups 9 or 10. Plasmid group 12 also contained smaller plasmids from *C. coli* and *C. jejuni* but were recovered from a broader range of hosts. In addition to human sources, these plasmids were recovered from *Campylobacter* isolates of cattle, wild-bird and unknown origin.

Plasmid group 18 differed from the other plasmid groups not belonging to the four major plasmid groups in group size, sequence length, plasmid host and isolation source. First, plasmid group 18 was the largest of these smaller plasmid groups and contained 17 plasmids that ranged between 25 and 73 kb in length. No plasmids from this group were recovered from *C. jejuni* or *C. coli*, and none were recovered from human or poultry sources. Rather, 15 plasmids were recovered from *C. fetus* of cattle origin, while the remaining two were recovered from the recently named *C. vicugnae* from goat and alpaca sources. Pangenomic clustering of the plasmid loci revealed that group 18 could be divided into 3 subgroups (Appendix A). The two plasmids from *C. vicugnae* were assigned to the same subgroup as other plasmids from *C. fetus* (Appendix A), indicating that the plasmids are shared between these two closely related *Campylobacter* species. Plasmids from group 18 encoded two variants of the T4SS (TXSScan: pT4SSt, CONJScan: dCONJ_typeT) and aligned to the pCC42 reference plasmid with up to 48.3% identity (Appendix A).

## 4. Discussion

Our pangenomic plasmid typing approach applied to a dataset of closed *Campylobacter* plasmid sequences yielded three key findings. First, the method successfully reconstructed the four primary *Campylobacter* plasmid types (pTet, pVir, pCC42, and pT6SS) without prior classification. This independent reconstruction of *Campylobacter* plasmid types demonstrates the method’s utility to identify conserved groups of related plasmids from sequence data alone. Second, our analysis revealed that the major plasmid groups were represented by multiple, genetically distinct plasmid subgroups. These plasmid subgroups shared the set of core loci representative of their plasmid group (e.g., pTet), along with a set of accessory loci that were unique to their specific subgroup (e.g., pTet.2) (Appendix A). Third, the identification of both group- and subgroup-specific loci offers a new framework for enhancing source attribution and AMR risk assessment in *Campylobacter* epidemiology.

The use of a single characterization method for all *Campylobacter* plasmids allows for enhanced comparisons between plasmid types. Prior to this investigation, similarities between *Campylobacter* plasmid types were identified by querying shared genes of interest (e.g., replicon, AMR or mobilization genes) or comparing the entire genetic content of differing plasmid types [25,32,51,52,62]. Our approach provides a unified system to compare all *Campylobacter* plasmids based on shared and exclusive loci [34]. This loci-based approach enables researchers to distinguish between plasmids that have acquired AMR or virulence genes via horizontal gene transfer versus those formed through plasmid fusion events. For example, the pVir.3 plasmids pN19C101.2, pN17C406.2 and pN58084.2 all encoded the hallmark *tet(O)* gene of the pTet plasmids. However, they contained only two pTet-specific typing loci, pTet49 and its neighboring intergenic region, both immediately upstream of *tet(O)* (Figure 4). This indicates that the integration of *tet(O)* into the pVir plasmids was a localized event and not the hybridization of two separate plasmids. This contrasts with the 14 *tet(O)*^+^ pT6SS plasmids where each plasmid contained 26–32 of the 38 pTet typing loci (Figure 5). The difference in pTet typing loci between *tet(O)*^+^ pVir plasmids and *tet(O)*^+^ pT6SS plasmids illustrates the method’s usefulness to infer evolutionary routes of gene acquisition.

This work bridges the gap between existing plasmid characterization and plasmid classification methods. Existing *Campylobacter* plasmid characterization approaches identify core and accessory genes present in specific plasmid types [25,63,64]. This approach produces a large pool of targets that accurately describe plasmid molecules. However, the pool of genes often contained elements present in other plasmid types. For example, the T4SS identified in the core genome of pTet plasmids is present in other plasmid types [25]. In contrast, methods to classify *Campylobacter* plasmids typically employ a small number of highly predictive loci to determine the plasmid type but are unable to describe the genetic diversity of plasmids within a given plasmid type [63]. One exception to this is the MOB-suite toolset that uses a combination of sequence alignment and biomarker matching to classify plasmids based on their similarity to a reference plasmid set [51,52]. While MOBTyper classifies *Campylobacter* plasmid types with similar resolution to the pangenomic method described here, the method prioritizes biomarkers associated with plasmid replication and transfer and may not include other medically relevant targets [51,52]. The pangenomic clustering of plasmids provided by the method used here not only provides a gene target-based plasmid classification method with high discriminatory power for plasmid subgroup identification, but it also organizes the data for downstream applications that include plasmid sequence typing and detection of structural rearrangements.

Analysis of *Campylobacter* plasmid sequences revealed that the plasmid groups could be divided into subgroups using subgroup-specific typing loci. These subgroups often displayed a bias for a given *Campylobacter* species, consistent with what has been described for the pCC42 group [32]. For example, among the pTet subgroups, only pTet.3 was prevalent in both *C. jejuni* and *C. coli*, suggesting possible cross-species transmission or shared ecological niches. Further, when multiple *Camplyobacter* species were represented in a plasmid subgroup, plasmids from different species could display differing AMR gene profiles. These subgroup-specific loci provide actionable targets for pathogen monitoring systems such as NARMS, allowing for early detection of high-risk plasmid lineages linked to AMR carriage. Finally, the integration of these typing loci into routine molecular surveillance workflows could automate the detection of specific plasmid subgroups. For example, the presence of the gene encoding the hypothetical protein A0A2U2XA67 from pTet.3 has the potential to indicate a plasmid with a greater likelihood of AMR gene carriage than a plasmid bearing the gene encoding the EexN family lipoprotein from pTet.2. The incorporation of plasmid typing loci into this analysis would expedite risk communication during pathogen investigations.

While the plasmid typing metrics identified in this study can be used to characterize plasmids in the sample dataset, our study has limitations that may affect its usefulness for analyzing the broader population of *Campylobacter* plasmids. First, we recognize the potential for sampling bias in our reference dataset. Plasmids from medically relevant *Campylobacter* isolates are more likely to be selected for long read sequencing than those collected through routine monitoring. Out of 15 *Campylobacter* species in our combined database, 87% of the plasmids were recovered from *C. coli* or *C. jejuni*. Further, 74% of the plasmids belonged to only four of the 30 plasmid groups. This limits applicability of conclusions drawn from the analyses of this dataset to less-studied *Campylobacter* species. Second, the definitions of the plasmid loci may be influenced by the annotation database used. The PROKKA annotation program excels at identifying many of the genes and regulatory elements found in the prokaryotic genome [46]. However, the correct identification of discipline-specific elements such as AMR or virulence genes can require the use of specialized databases. Our analysis incorporated directed searches for AMR, virulence and bacterial secretion system elements, but did not include an investigation of IS, transposon or CRISPR elements. The addition of these or other discipline-specific markers may improve the relevance of the *Campylobacter* plasmid definitions.

In summary, our findings demonstrate the value of using a single, scalable pangenomic framework for *Campylobacter* plasmid typing. Our approach was not only able to recapitulate the four major *Campylobacter* plasmid groups, it was also able to further subtype them into distinct lineages of plasmid subgroups. The enhanced characterization provided by this method revealed associations between the plasmid subgroup with both AMR gene carriage and host species. This multilocus plasmid typing approach enables researchers to not only detect plasmids from the major plasmid groups but also identify the specific subgroup the plasmid belongs to. As a result, application of the Lociq method to *Campylobacter* plasmids can support enhanced source attribution and inform risk assessment evaluations, supporting efforts to protect human and animal health.

## Figures and Tables

**Figure 1 pathogens-14-00936-f001:**
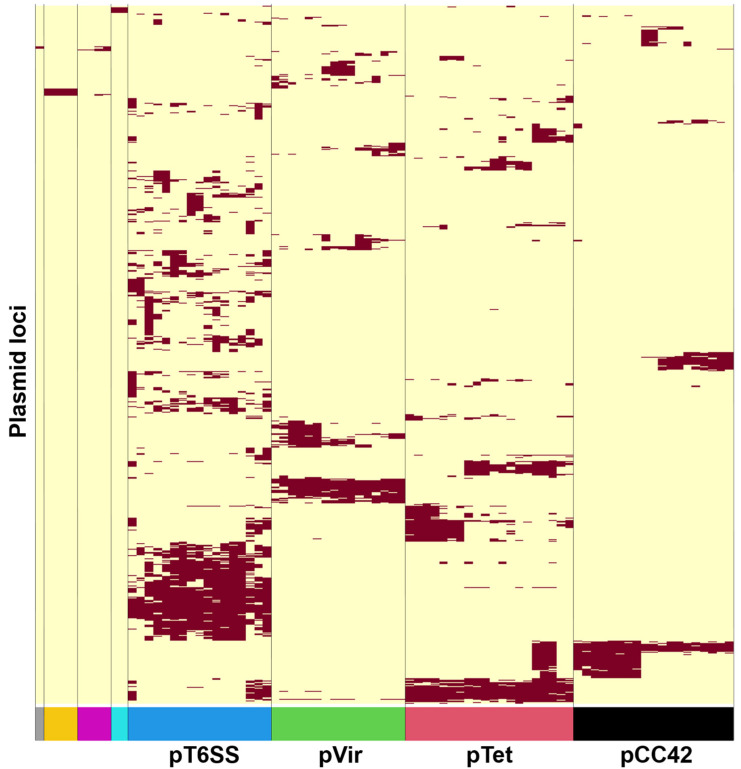
Visual representation of the pangenome of 83 NARMS *Campylobacter* plasmids where plasmids are organized along columns and loci among rows and cells indicate the presence or absence of loci through a dark red or light-yellow color, respectively. Color bars on the bottom correspond to *Campylobacter* plasmid groups and the four largest groups are labeled as the four major *Campylobacter* plasmid types.

**Figure 2 pathogens-14-00936-f002:**
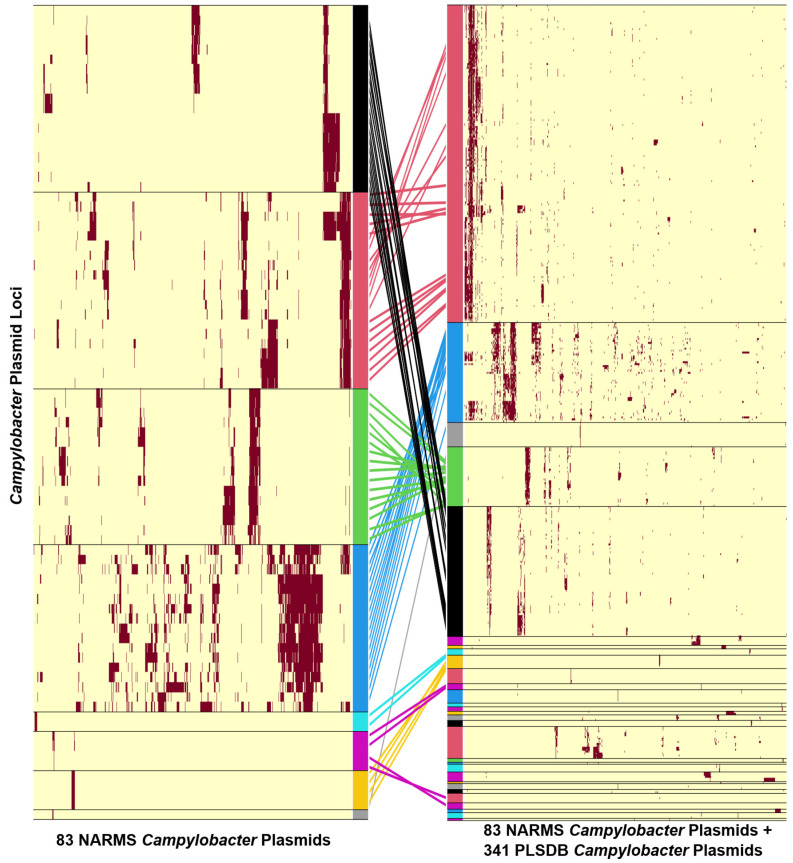
Visual representation of two *Campylobacter* plasmid pangenomes: the 83 plasmids from NARMS isolates (**left**) and the combined dataset of 424 plasmids (**right**). Color bars correspond to the same *Campylobacter* plasmid groups as Figure 1. Plasmids from the NARMS pangenome are connected to their positions in the combined pangenome through lines color-coded to their plasmid group.

**Figure 3 pathogens-14-00936-f003:**
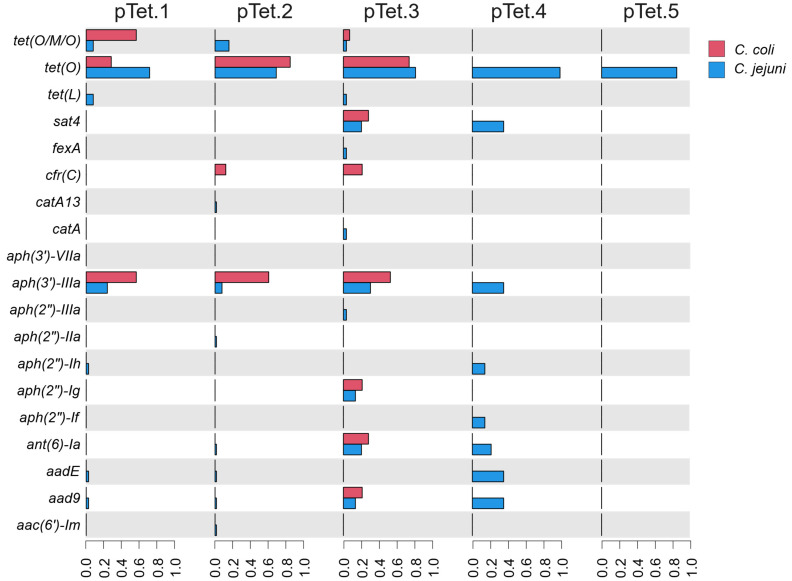
Bar graph showing the prevalence of AMR genes among the *Campylobacter* pTet plasmid subgroups where the x-axis details the proportion of isolates within the subgroup bearing the corresponding AMR gene. *C. coli* is represented with red bars while *C. jeuni* is represented with blue bars.

**Figure 4 pathogens-14-00936-f004:**
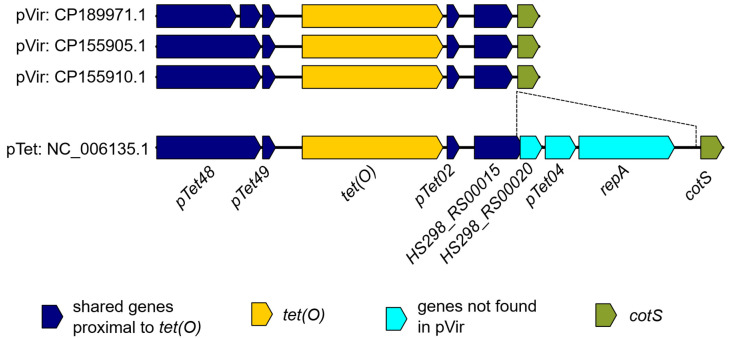
Comparison of the *tet*(*O*) region of the reference NC_006135.1 pTet plasmid and the pVir.3 *tet*(*O*)^+^ plasmids in the dataset. Genes shared between the four plasmids are in blue; the *tet*(*O*) gene is in yellow; the genes not found in pVir are in cyan; and the shared *cotS* gene downstream of *tet*(*O*) is in green.

**Figure 5 pathogens-14-00936-f005:**
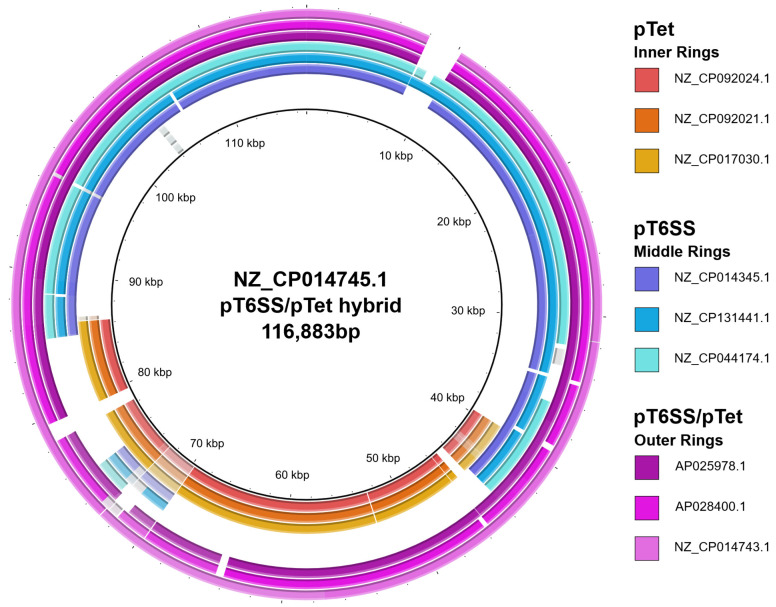
Alignment of pTet, pT6SS and hybrid pTet/pT6SS plasmids to a hybrid pT6SS/pTet reference. Three *Campylobacter* pTet sequences (inner three alignments, orange), non-hybrid pT6SS sequences (middle three alignments, blue) and hybrid pT6SS/pTet sequences (outer three alignments, purple) were mapped to the NZ_CP014745.1 hybrid pT6SS/pTet plasmid reference sequence to identify the pTet and pT6SS regions of the hybrid plasmid. This image was generated using the Blast Ring Image Generator (BRIG) v0.95 [61].

**Figure 6 pathogens-14-00936-f006:**
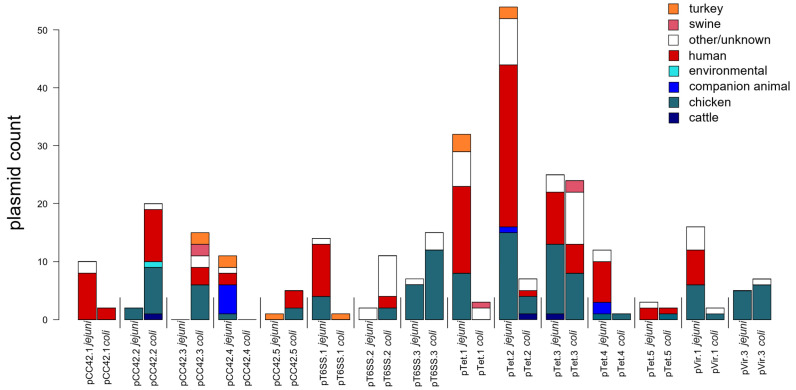
Source attribution bar graph of plasmid subgroups from the major *Campylobacter* species *C. coli* and *C. jejuni*. Bar height corresponds to the number of plasmids isolated from a given plasmid subgroup.

**Table 1 pathogens-14-00936-t001:** Alignment of *Campylobacter* Plasmid Sequences to Reference Sequence.

Plasmid Group ID	Combined Dataset (n)	pTet Mean % ID	pVir Mean % ID	pCC42 Mean % ID	pT6SS Mean % ID
pTet	165	90.9	0.5	2.4	37.4
pVir	31	2.0	89.0	0.0	0.8
pCC42	67	0.9	0.2	90.0	0.4
pT6SS	52	36.1	0.3	4.5	65.2
26 minor groups	109	2.6	0.1	4.9	1.2

Reference plasmid sequences are NC_006135.1 (pTet), AF226280.2 (pVir), MH634987.1 (pCC42) and CP014743.1 (pT6SS) and values shown represent the mean alignment value for all plasmids of a plasmid group to the reference plasmid.

**Table 2 pathogens-14-00936-t002:** pCC42 Typing Loci Identity by Subgroup.

Typing Locus	Subgroup Variant 1	Subgroup Variant 2	% Identity Between Variants 1 and 2
TrbE	pCC42.2	pCC42.1, pCC42.4	91.72
TrbH	pCC42.4	pCC42.2, pCC42.3	79.31
TrbD	pCC42.5	pCC42.2	82.09
TraL	pCC42.4	pCC42.2, pCC42.5	87.67
Signal Peptidase I	pCC42.4	pCC42.2	87.1
Uniprot: S3X8X6	pCC42.2, pCC42.3	pCC42.1, pCC42.4, pCC42.5	91.3
Uniprot: S3YFQ1	pCC42.2	pCC42.1, pCC42.4	92.06

## Data Availability

Accession numbers for all isolates may be found in the first table of the Appendix A.

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
