# Peer review of "Pangenomic Characterization of Campylobacter Plasmids for Enhanced Molecular Typing, Risk Assessment and Source Attribution"

_pathogens, 2025, doi:10.3390/pathogens14090936_

Round 1

Reviewer 1 Report

Comments and Suggestions for Authors

In this manuscript the authors are trying to create a Campylobacter plasmid typing approach using pangenomic analysis of a number of plasmid sequences. However there is a potential for sampling bias in the reference dataset they used and the majority of plasmids belonged to only four of the 30 plasmid groups they produced. There proposed typing method might not be applicable to Campylobacter species with less available plasmid sequences. I have listed few comments/concerns below.

A. Major Comments/Concerns:

The major concern I have is the discussion section. There is no single citation in the whole discussion. Citations are crucial to support claims in scientific writing even if the authors are using a novel approach. There are several previous relevant studies some of them were cited by the authors in the introduction section but are completely absent from the discussion section. These previous trials and their conclusions should be mentioned in the discussion section. This is probably the first discussion I read in published papers or reviewed as manuscripts that lack a single citation. I suggest that the authors include relevant citations to compare their method to the previous plasmid classification or typing trials to put their conclusion in context of previous work. 

B. Minor Comments:

  1. Page 2 line 82 - 83. Reference 27 is about a 119 kb megaplasmid involved in blood cell lysing and not about 190 kb plasmid involved in adhesion during infection. Did you mean reference 37? Please correct.
  2. Page 4 line 155 - 156. The sentence "The remaining 4 groups ................................. ranged in size between 2-4 plasmids each" is missing something and does not make sense. Please rewrite for clarity. 

Author Response

I. COMMENTS

In this manuscript the authors are trying to create a Campylobacter plasmid typing approach using pangenomic analysis of a number of plasmid sequences. However there is a potential for sampling bias in the reference dataset they used and the majority of plasmids belonged to only four of the 30 plasmid groups they produced. There proposed typing method might not be applicable to Campylobacter species with less available plasmid sequences. I have listed few comments/concerns below.

  1. Major Comments/Concerns:

The major concern I have is the discussion section. There is no single citation in the whole discussion. Citations are crucial to support claims in scientific writing even if the authors are using a novel approach. There are several previous relevant studies some of them were cited by the authors in the introduction section but are completely absent from the discussion section. These previous trials and their conclusions should be mentioned in the discussion section. This is probably the first discussion I read in published papers or reviewed as manuscripts that lack a single citation. I suggest that the authors include relevant citations to compare their method to the previous plasmid classification or typing trials to put their conclusion in context of previous work.

I. RESPONSE

We agree that much excellent work has gone into the characterization of Campylobacter plasmids and many of the loci identified in this study are consistent with the core genes of Campylobacter plasmids identified elsewhere.  The key difference, however, is that our final loci list filters out genes that are present above a threshold value outside of a given plasmid group.   What this means is that while other works have presented the backbone region of Campylobacter plasmids, we show which regions of the backbone demonstrate group exclusivity and are indicative of a given plasmid type.  We then take this a step further to identify the specific loci that demarcate potential lineages within a given plasmid group.   This has the secondary benefit of providing multiple genetic targets to screen for highly-fragmented assemblies where other plasmid typing schema may be unable to resolve the plasmid type.  We have added this topic as well as supporting references into the Discussion.

We also agree that an acknowledgement of other plasmid typing schema should be included.  This was not included in the initial draft to prevent misleading readers into an expectation of a method comparison paper.  Additionally, this manuscript does not contain the complete plasmid-level characterization (including individual plasmid sequence type and typing by sequence-rearrangement events) provided by the Lociq plasmid typing approach, so any comparisons to other methods would not accurately reflect its discriminatory power.  However, we agree with the reviewer that our results should be viewed in the context of existing typing methods.  As such, we have added the primary and secondary cluster IDs from MOB-Typer to Supplemental Table 1, and added a Supplemental Figure showing the alignment of plasmid classification from the MOB-Typer secondary cluster ID to the pangenomic plasmid subgroups identified in this study (please see attached file).  The alignment shows that even before we apply the Lociq sequence typing or sequence-rearrangement typing results, the plasmid subgroupings identified in this study correlate with the MOB-Typer secondary cluster IDs.

  1. Minor Comments:

II. Page 2 line 82 - 83. Reference 27 is about a 119 kb megaplasmid involved in blood cell lysing and not about 190 kb plasmid involved in adhesion during infection. Did you mean reference 37? Please correct.

II. We agree with the reviewer that this should be changed.  The 190kb size mentioned was in reference to the largest pT6SS plasmid in our dataset and the adhesion that was mentioned was mistakenly taken from the contribution of the T6SS to virulence, not the role of the plasmid.  We have updated the text to reflect the appropriate plasmid size and red blood cell cytotoxicity in described in the reference.

III. Page 4 line 155 - 156. The sentence "The remaining 4 groups ................................. ranged in size between 2-4 plasmids each" is missing something and does not make sense. Please rewrite for clarity.

III. We have reworded this section for clarity to read:  "As our initial analysis identified 8 plasmid groups, the remaining 4 plasmid groups contained the 11 plasmids left in the dataset."

Reviewer 2 Report

Comments and Suggestions for Authors

In this study, Harrison et al. devised a typing scheme relying on pangenome analysis that classifies Campylobacter plasmids with a greater discriminatory power, which could be valuable as a molecular surveillance tool. The manuscript generally reads well but could be improved further by addressing the following issues.
1. In the Results, cite tables and figures more frequently and further highlight major findings. In lines 298-317, use the past tense when describing the data. 
2 In the Materials and Methods, add extra information for the analysis process as mentioned in specific comments and specify whether any parameters for software were changed from the default values. 
3. References are often missing through the manuscript. Add references whenever necessary.
4. Revise unclear statements pinpointed in the specific comments to improve readability. 
5. Remove the title of figures within the images. 

Specific comments:
- Lines 54 and 231: Remove the colon. 
- Lines 100-102: Re-write this statement for clarity and provide further information for the Campylobacter sequences. Given that 58 strains were sequenced for this study, 25 sequences were retrieved from NCBI? Which programs were used to retrieve sequences from NCBI and PLSDB?
- Line 103: Show the full name for NARMS instead of line 150. 
- Lines 104 and 106: Show the geographical location (city and country) of the manufacturer. 
- Line 105: Add a comma after "version MIN-101B". 
- Line 108: Specify what "support" implies. 
- Line 110: Change "then" to "and then". 
- Line 113: I believe that "Plasmid sequences" should be changed to "Genome sequences" given that genomic DNA was sequenced. 
- Line 131: What is "31" in "Lociq31"?
- Lines 133-140: This seems to be redundant with lines 128-132.
- Line 135: Change "two-stage" to "two-step". 
- Lines 144-146: Which programs were used to query the protein sequences? 
- Lines 146 and 147: Specify which program in EMBOSS was used. 
- Lines 156-158: This statement could be incorporated into the statement in lines 152 and 153. 
- Lines 161 and 162: "341 plasmid sequences ... species" should also have been mentioned in the Materials and Methods. 
- Line 167: Throughout the manuscript, change "aligned to" to "were/was aligned to". 
- Line 178: Include the program to be used to create elbow plots in the Materials and Methods. 
- Line 190: Change "per loci" to "per locus". 
- Line 194: Change "encoded for" to "encoded" throughout the manuscript.
- Line 195: Double-check "type T T4SS genes". 
- Line 214: Change "proteins with" to "genes encoding proteins with". 
- Lines 224 and 225: Revise this statement for clarity. 
- Lines 239 and 240: Revise this statement for clarity. 
- Line 246: Change "this this" to "this". 
- Lines 247 and 248: Change "cassette flanking" to "cassette-flanking". 
- Line 254: I believe that "homolog" should be removed. 
- Line 266: Revise "human, avian and a single ursine source". 
- Lines 274 and 275: Revise this statement for clarity.
- Lines 290 and 291: Revise this statement for clarity.
- Line 299: Change "9 & 10" to "9 and 10". 
- Line 308: In "smaller plasmid groups", smaller than what?
- Line 318: Please remove this heading. 
- Line 342: Change "based" to "based on". 
- Line 346: Change "pTet specific" to "pTet-specific". 
- Lines 364 and 365: Revise "previously ... plasmid sequences" for clarity. 

Table 1
- It could be informative if the authors include the variance of the alignment values. 

Figure 1
- Change "A visual representation the" to "Visual representation of the". 

Figure 2
- Change "A visual representation" to "Visual representation". 

Figure 3
- Change "A bar graph" to "Bar graph". 
- Change "C. coli are" and "C. jejuni are" to "C. coli is" and "C. jejuni is", respectively. 
- Explain the x-axis in the text. 

Figure 4
- Change "A comparison of" to "Comparison of". 
- Italicize "cotS". 

Figure 5
- Show which program was used to create this figure. 

Figure 6
- Change "representation for" to "representation of".

Author Response

In this study, Harrison et al. devised a typing scheme relying on pangenome analysis that classifies Campylobacter plasmids with a greater discriminatory power, which could be valuable as a molecular surveillance tool. The manuscript generally reads well but could be improved further by addressing the following issues.

Comment 1. In the Results, cite tables and figures more frequently and further highlight major findings. In lines 298-317, use the past tense when describing the data.

Response 1. We agree that more frequent references to the supporting figures would be beneficial to the reader and have added these to the manuscript.  Additionally reworded the selected section to the appropriate text.

Comment 2. In the Materials and Methods, add extra information for the analysis process as mentioned in specific comments and specify whether any parameters for software were changed from the default values.

Response 2. The Materials and Methods section has been updated to include references to the additional parameters used when the default settings were not selected, as well as the appropriate citations for the software used.

Comment 3. References are often missing through the manuscript. Add references whenever necessary.

Response 3. We have updated the manuscript to include additional references, as well as cite existing references multiple times where appropriate.

Comment 4. Revise unclear statements pinpointed in the specific comments to improve readability.

Response 4. We have revised the selected statements for clarity, please see responses to the specific comments for details.

Comment 5. Remove the title of figures within the images.

Response 5. Titles have been removed from the figures

Specific comments:

Comment 6. - Lines 54 and 231: Remove the colon.

Response 6. The colon has been removed.

Comment 7. - Lines 100-102: Re-write this statement for clarity and provide further information for the Campylobacter sequences. Given that 58 strains were sequenced for this study, 25 sequences were retrieved from NCBI? Which programs were used to retrieve sequences from NCBI and PLSDB?

Response 7. We agree that the sentence is unclear.  Briefly, the NARMS dataset was combination of two sources.  First, there are the sequences that were generated onsite specifically for this study.  The second source was NARMS isolates that had previously been uploaded to NCBI, and the sequences were downloaded through the web interface using a list of accession IDs.  No non-NARMS Campylobacter plasmid sequences were recovered directly from NCBI.  The non-NARMS Campylobacter plasmids  in NCBI would be accounted for in the curated PLSDB database and are represented in that dataset.

We have reworded the sentence to read:

“Plasmid sequences were recovered from two sources.  The dataset of Campylobacter plasmids from the US National Antimicrobial Resistance Monitoring System (NARMS) used in this study was represented by a combination of plasmids that were sequenced onsite for this study and closed Campylobacter plasmid sequences from the NARMS program that had been previously uploaded to the National Center for Biotechnology Information NCBI [32].  The dataset of 341 Campylobacter plasmids from PLSDB were recovered by downloading a local copy of the PLSDB v2023_11_03_v2 [33] database and extracting all sequences from entries associated with a Campylobacter isolation source.”

We have also added a column to Table S1 to differentiate NARMS from PLSDB plasmids.

Comment 8. - Line 103: Show the full name for NARMS instead of line 150.

Response 8. We have changed this instance to the National Antimicrobial Resistance Monitoring System and removed it from its latter mention.

Comment 9.- Lines 104 and 106: Show the geographical location (city and country) of the manufacturer.

Response 9. Geographical location information of the manufacturers has been added.

Comment 10.- Line 105: Add a comma after "version MIN-101B".

Response 10. We have added a comma to close the clause in the sentence.

Comment 11.- Line 108: Specify what "support" implies.

Response 11. The Trycycler workflow merges replicates of assemblies from multiple sequence assemblers with the intent of leveraging the strengths of each to create an improved final assembly.  In this case, the assemblers used in the workflow were FLYE, Raven and miniasm.

The sentence has been modified to read: "Sequences were assembled with Trycycler v0.5.5 using FLYE v2.9.3, Raven v1.8.3 and miniasm v0.3-r179 to generate the initial assemblies."

Comment 12.- Line 110: Change "then" to "and then".

Response 12. We have modified the sentence accordingly.

Comment 13.- Line 113: I believe that "Plasmid sequences" should be changed to "Genome sequences" given that genomic DNA was sequenced.

Response 13. We agree that language specificity is important here.  As such, we have changed the wording to “Plasmid assemblies” to reflect that it was the plasmid assemblies that were annotated with PROKKA.  Prior to this, the genomic portion of the assemblies representing the ~1.6-1.8mb contig was removed and not subjected to annotation.

Comment 14.- Line 131: What is "31" in "Lociq31"?

Response 14. '31' is a formatting error in the citation that occurred when transferring the original text to the journal template.  We have reformatted the sentence to correct this error.

Comment 15. - Lines 133-140: This seems to be redundant with lines 128-132.

Response 15. We agree that the processes are very similar, but they apply to two separate stages.  Lines 128-132 describe the process of identifying the major plasmid groups from the pangenome.  Lines 133-140 describe the process of subdividing a selected major plasmid group.  In the first stage, the within-group and without-group loci filtering steps are applied to the same subset of data, the plasmid group of interest.  In the second stage, the without-group filtering is applied to entire plasmid group of interest, while the within-group filtering is applied to the specific subgroup of the plasmid group of interest.

We have changed the wording to read: “were determined in a similar two-step process” to inform the reader that the description of the two processes differs slightly.

Comment 16. - Line 135: Change "two-stage" to "two-step".

Response 16. We have modified the sentence to include “two-step”

Comment 17. - Lines 144-146: Which programs were used to query the protein sequences?

Response 17. The protein sequences were queried against the databases using the Uniprot and KEGG web interfaces through a multifasta upload.  We have updated the text to reflect this.

Comment 18. - Lines 146 and 147: Specify which program in EMBOSS was used.

Response 18.0We used the cons program within EMBOSS to generate the consensus sequences using default parameters and have added this information to the text.

Comment 19. - Lines 156-158: This statement could be incorporated into the statement in lines 152 and 153.

Response 19. The initial placement of the sentence was intended to highlight the overrepresentation of sequences in our dataset within the 4 plasmid groups.  We have added the relevant information from 156-158 to 152-153.

Comment 20. - Lines 161 and 162: "341 plasmid sequences ... species" should also have been mentioned in the Materials and Methods.

Response 20. We have added the following to line 104 in the Materials and Methods: "…to generate a dataset of 424 plasmid sequences from 15 Campylobacter species.         

Comment 21. - Line 167: Throughout the manuscript, change "aligned to" to "were/was aligned to".

Response 21. This change has been made.

Comment 22. - Line 178: Include the program to be used to create elbow plots in the Materials and Methods.

Response 22. Elbow plots were generated in R without the use of any external libraries.  Figure generation information has been added to the Materials and Methods.

Comment 23. - Line 190: Change "per loci" to "per locus".

Response 23. We have changed the plural to singular.

Comment 24. - Line 194: Change "encoded for" to "encoded" throughout the manuscript.

Response 24. This change has been made.

Comment 25. - Line 195: Double-check "type T T4SS genes".

Response 25. We agree that "type T T4SS" appears to be a typo, however these genes refer to the T4SS_typeT output of the CONJScan module.

Comment 26. - Line 214: Change "proteins with" to "genes encoding proteins with".

Response 26. We have modified the sentence to read: "...the presence of genes encoding proteins with amino acid sequence similarity to..." to state that we identified the genes, not the proteins, and also that the sequence comparisons were performed on the amino acid sequences, not the nucleotides.

Comment 27. - Lines 224 and 225: Revise this statement for clarity.

Response 27. We have revised the statement to read: "Analysis of protein-coding loci among the pCC42 subgroups showed that seven loci pairs shared more than 70% of their sequence."

Comment 28. - Lines 239 and 240: Revise this statement for clarity.

Response 28. The sentence has been modified to read: "A 4,904bp sequence surrounding tet(O) was conserved in all pVir tet(O)+ plasmids. The aligned region of this sequence had >99% sequence identity to a 7,952bp region of the reference pTet plasmid.  While the aligned region contained both tet(O) and repA, repA was missing from the pVir tet(O)+ plasmids."

Comment 29. - Line 246: Change "this this" to "this".

Response 29. We have removed the redundant “this”.

Comment 30. - Lines 247 and 248: Change "cassette flanking" to "cassette-flanking".

Response 30. We agree that adding a hyphen would improve clarity, but applied it elsewhere in phrase: gene-cassette flanking regions.

Comment 31. - Line 254: I believe that "homolog" should be removed.

Response 31. We agree that the wording describing the evaluation of sequence similarity to a homolog appears redundant, however 'stage 0 sporulation protein A homolog' is the title of the closest sequence match in the Uniprot databse.  We have added the protein IDs in this section to specify the similarity to a specific record rather than the similarity to general family of homologous sequences.

Comment 32. - Line 266: Revise "human, avian and a single ursine source".

Response 32. For consistency, we have modified the sentence to read: "These pT6SS plasmids were recovered from Campylobacter strains isolated from human sources, avian sources and a single ursine source"

Comment 33. - Lines 274 and 275: Revise this statement for clarity.

Response 33. We have reworded the sentence to read: "Additionally, in comparison to a 96.4% carriage rate of tet(O) in pTet plasmids, only 87.5% of the hybrid pT6SS/pTet plasmids bore the tet(O) gene."

Comment 34. - Lines 290 and 291: Revise this statement for clarity.

Response 34. The original statement reads:

“In addition to the four largest groups described above (pTet, pVir, pCC42 and pT6SS), pangenomic clustering of plasmid loci produced 26 more plasmid groups”

we have revised it to the following:

“The pTet, pVir, pCC42 and pT6SS plasmid groups mentioned above represent four of the 30 plasmid groups that were identified.”

We recognize that summarizing the plasmid group identification process in this sentence as ‘pangenomic clustering of plasmid loci’ is not necessary.

Comment 35. - Line 299: Change "9 & 10" to "9 and 10".

Response 35. We have made this change.

Comment 36. - Line 308: In "smaller plasmid groups", smaller than what?

Response 36. This refers to the plasmid groups that were smaller than the four major Campylobacter plasmid groups.  For specificity, we have modified the sentence to read: "Plasmid group 18 differs from the other plasmid groups not belonging to the four major plasmid groups in group size, sequence length, plasmid host and isolation source."

Comment 37. - Line 318: Please remove this heading.

Response 37. The heading "3.2. Figures, Tables and Schemes" was part of the template document and retained for organization.  We have removed the heading in this version.

Comment 38. - Line 342: Change "based" to "based on".

Response 38. We have made this change to the sentence.

Comment 39. - Line 346: Change "pTet specific" to "pTet-specific".

Response 39. We have made this change to the sentence.

Comment 40. - Lines 364 and 365: Revise "previously ... plasmid sequences" for clarity.

Response 40. We have revised this section to read: "Finally, the integration of these typing loci into routine molecular surveillance workflows could automate the detection of specific plasmid subgroups.   For example, the presence of the gene encoding the hypothetical protein A0A2U2XA67 from pTet.3 has the potential to indicate a plasmid with a greater likelihood of AMR gene carriage than a plasmid bearing the gene encoding the EexN family lipoprotein from pTet.2.  The incorporation of these typing loci into this analysis would expedite risk communication during pathogen investigations."

Comment 41. Table 1

- It could be informative if the authors include the variance of the alignment values.

Response 41. We agree that this information would be useful, and columns D:K of Supplemental Table 1 contain the alignment values of each plasmid to the reference plasmid from the four major plasmid groups.

Comment 42. Figure 1

- Change "A visual representation the" to "Visual representation of the".

Response 42. We have made this change.

Comment 43. Figure 2

- Change "A visual representation" to "Visual representation".

Response 43. We have made this change.

Comment 44. Figure 3

- Change "A bar graph" to "Bar graph".

- Change "C. coli are" and "C. jejuni are" to "C. coli is" and "C. jejuni is", respectively.

- Explain the x-axis in the text.

Response 44. We have made the requested changes and updated the text to read:

“Bar graph showing the prevalence of AMR genes among the Campylobacter pTet plasmid subgroups where the x-axis details the percentage of isolates within the subgroup bearing the corresponding AMR gene.  C.  coli is represented with red bars while C.  jeuni is represented with blue bars.  Plasmids isolated from C.  coli were only represented in subgroups 1, 2 and 3.”

Comment 45. Figure 4

- Change "A comparison of" to "Comparison of".

- Italicize "cotS".

Response 45. We have made this change.

Comment 46. Figure 5

- Show which program was used to create this figure.

Response 46. We have added the following to the figure legend:

“Image was generated using the Blast Ring Image Generator (BRIG) v0.95.”

Comment 47. Figure 6

- Change "representation for" to "representation of".

Response 47. We have made this change.

Round 2

Reviewer 1 Report

Comments and Suggestions for Authors

I am fine with the revised version of the manuscript.

Author Response

Comment 1: I am fine with the revised version of the manuscript.

Response 1: We thank the reviewer for their work and their time.

Reviewer 2 Report

Comments and Suggestions for Authors

Many comments and concerns raised in the previous review have been addressed in the revised manuscript, which improved considerably compared with the original submission. Nonetheless, the following issues still remain unresolved.

  1. Many statements in the Results section still need citations of proper tables and figures. Some (not all) examples are shown in the specific comments.
  2. Several statements in the revised manuscript should be further modified to enhance clarity. Please see specific comments.
  3. Line 119: In the rebuttal, the authors said that the chromosomal sequences were eliminated before annotation. Include the procedure to remove the chromosomal contigs.
  4. Lines 330-332: In Table S1, two plasmids of group 18 were marked with "0" and the other group 18 plasmids were marked with "1" in the pT4SS column. Explain how this means "two variants of the T4SS". Also, in Table S1, any percent identity is not included.

Specific comments:

- Line 36: Change "highly-clonal" to "highly clonal".

- Line 43: Show the full name for "WGS".

- Line 44: Change "clinically-relevant" to "clinically relevant".

- Line 46: Change "schema" to "scheme".

- Line 83: Change "cell specific" to "cell-specific".

- Lines 83-85: Provide references.

- Lines 100-105: I recommend including the number of all the NARMS plasmids as well as those of the plasmids sequenced for this study and the closed plasmid sequences retrieved from the NCBI database.

- Line 108: Change "collected" to "isolated".

- Lines 109 and 110: Change "Whole genome sequencing (WGS)" to "WGS".

- Line 114: Is "mb" a typo for "Mb"?

- Line 152: Change "typing loci set" to "typing loci".

- Line 170: Move "(Figure S2)" to the end of the statement.

- Line 175: In Table 1, the number for the pTet reference plasmid aligned to the pT6SS plasmid sequences is 37.4. Please change "36%" to "37%". For lines 175-177, cite Table 1.

- Lines 187 and 188: Cite Figure S3.

- Lines 191 and 192: Re-write this statement for clarity.

- Line 194: Move "(Table S3)" to the end of the statement.

- Line 202: Throughout the manuscript, change "encoded for" to "encoded". This suggestion was already mentioned in the previous review but was not fully addressed.

- Lines 202 and 203: Clarify "T4SS modified for protein transfer".

- Line 210: Change "identified each in" to "identified in each of".

- Line 213: Change "than in" to "than".

- Lines 236 and 252: Clarify "protein modified T4SS".

- Lines 238-240: I recommend removing "were found among the pCC42 subgroup", which is redundant.

- Lines 240-242: Re-write this statement for clarity.

- Lines 256-260: Re-write these statements for clarity.

- Line 277: Change "compared to" to "compared with".

- Lines 284 and 285: I cannot find "hybrid pT6SS/pTet plasmids"  in Table S1. Explain this.

- Lines 290 and 291: Revise this statement for clarity.

- Lines 300-302: This statement is confusing since "the respective ... organisms" has nothing to do with isolation source. Revise it.

- Lines 304 and 305: Re-write this statement for clarity.

- Line 310: Change "four" to "four major groups".

- Line 312: Change "greater than" to "more than".

- Line 359: I recommend eliminating "(HGT)" since this acronym is not used again.

- Line 361: Change "encode the hallmark" to "encoded the hallmark".

- Lines 362 and 363: Change "contain" to "contained".

- Line 385: Change "it also organizes" to "but it also organizes".

- Lines 390 and 391: Change "has been previously described" to "has previously been described".

- Figure 1. I cannot find "color bars on the top". "top" should be replaced with "bottom".

- Figure 2. Change "to their position" to "to their positions".

- Figure 3. I recommend eliminating "Plasmids isolated ... and 3". Change "percentage" to "ratio".

- Figure 4: Change "A comparison" to "Comparison". This was mentioned in the previous review but has not been addressed.

- Figure 5: Change "Image was" to "The image was".

- Figure 6: Change "Bar graph representation" to "Bar graph".

Author Response

Comment 1: Many statements in the Results section still need citations of proper tables and figures. Some (not all) examples are shown in the specific comments.

Response 1: We have included additional references to figures and tables in the manuscript.  However, to simplify the text, we also follow the convention of discussing a figure or table when it is cited earlier in the paragraph, and there are no other references between the sentence and the figure in the same paragraph.

Comment 2:  Several statements in the revised manuscript should be further modified to enhance clarity. Please see specific comments.

Line 119: In the rebuttal, the authors said that the chromosomal sequences were eliminated before annotation. Include the procedure to remove the chromosomal contigs.

Response 2: The assemblies were generated from long read sequencing data and chromosomal contigs could be identified through length and completeness using the outputs from programs described in the methods section.  Contigs with a sequencing depth less than that of the chromosome were marked as unusable.  Removal of the chromosomal sequences involved a simple approach of splitting the multifasta files into multiple single fasta files using awk (awk approach:   cat "$1" | awk '{if (substr($0, 1, 1)==">") {filename=(substr($0,2)".fna")} print $0 > filename}') and selecting the files that were not identified as being of chromosomal origin from the polishing programs.  We have added the following line:

“Contigs larger than 1.4Mb were identified as chromosomal sequences, while smaller contigs with lower depth of coverage than the chromosomal contigs were removed.”

Comment 3: Lines 330-332: In Table S1, two plasmids of group 18 were marked with "0" and the other group 18 plasmids were marked with "1" in the pT4SS column. Explain how this means "two variants of the T4SS". Also, in Table S1, any percent identity is not included.

Response 3: In the sentence "Plasmids from group 18 encoded two variants of the T4SS and aligned to the pCC42 reference plasmid with up to 48.3% identity (Table S1)."  The section "aligned to the pCC42 reference plasmid with up to 48.3% identity" references columns H-O where the weighted percent identity values are contained.   Information regarding the "two variants of the T4SS" can be found in columns R and T of Table S1 referring to the TXSSCan result of the pT4SSt system and the CONJScan result of the dCONJ_typeT system.  Both of these are variants of a T4SS, and we have included these identifiers in the text to notify the reader which two variants were detected.

Specific comments:

Comment 4: - Line 36: Change "highly-clonal" to "highly clonal".

Response 4: We have made this change.

Comment 5: - Line 43: Show the full name for "WGS".

Response 5: We have made this change.

Comment 6: - Line 44: Change "clinically-relevant" to "clinically relevant".

Response 6: We have made this change.

Comment 7: - Line 46: Change "schema" to "scheme".

Response 7: The current revision contains "scheme" so we assume that the reviewer intended "schema" to replace "scheme".

Comment 8: - Line 83: Change "cell specific" to "cell-specific".

Response 8: We have made this change.

Comment 9: - Lines 83-85: Provide references.

Response 9: We have included a reference that shows that the accessory gene profiles of Campylobacter plasmids are not uniform among plasmid types.

Comment 10: - Lines 100-105: I recommend including the number of all the NARMS plasmids as well as those of the plasmids sequenced for this study and the closed plasmid sequences retrieved from the NCBI database.

Response 10: For clarity, all plasmids sequenced in-house for this study were NARMS plasmids.  Also, the only plasmids recovered from NCBI were the plasmids that had been previously sequenced and uploaded through the NARMS program.  In this way, all plasmids in our dataset that were sequenced for this study or recovered from NCBI were NARMS plasmids.

However, there are additional closed Campylobacter plasmid sequences in the NCBI database that were not generated through the NARMS program.  If these plasmid sequences are accounted for in PLSDB, they have been attributed as such in Table S1.  If they were not included in PLSDB, they would not have been included in this study.

For completeness, if within the set of plasmids from NCBI that were accounted for in PLSDB a duplicate of our onsite-sequenced NARMS plasmids was present, the duplicate record was removed and the original NARMS attribution was retained.

Comment 11: - Line 108: Change "collected" to "isolated".

Response 11: We have replaced this with "recovered" so the reader does not make a false connection between "isolates" and "isolated"

Comment 12: - Lines 109 and 110: Change "Whole genome sequencing (WGS)" to "WGS".

Response 12: We have made this change.

Comment 13: - Line 114: Is "mb" a typo for "Mb"?

Response 13: We have made this change.

Comment 14: - Line 152: Change "typing loci set" to "typing loci".

Response 14: This redundancy was removed.

Comment 15: - Line 170: Move "(Figure S2)" to the end of the statement.

Response 15: We have made this change.

Comment 16: - Line 175: In Table 1, the number for the pTet reference plasmid aligned to the pT6SS plasmid sequences is 37.4. Please change "36%" to "37%". For lines 175-177, cite Table 1.

Response 16: The 36% value is correct, at the intersection of pTet mean % ID column to pT6SS row.  We have changed the value in-text to 36.1% to help orient the reader.

Comment 17: - Lines 187 and 188: Cite Figure S3.

Response 17: Lines 187 and 188 describe the elbow plots mentioned in the previous sentence where Figure S3 was cited.

Comment 18: - Lines 191 and 192: Re-write this statement for clarity.

Response 18: We have added the adjusted Rand index value and re-written the statement as:

Typing of these subgroups showed a strong agreement with the secondary cluster ID assigned by the MOBTyper program (adjusted Rand index, 0.669) (Figure S4).

Methods and references to this have been updated accordingly.

Comment 19: - Line 194: Move "(Table S3)" to the end of the statement.

Response 19: We have made this change.

Comment 20: - Line 202: Throughout the manuscript, change "encoded for" to "encoded". This suggestion was already mentioned in the previous review but was not fully addressed.

Response 20: We have made this change.

Comment 21: - Lines 202 and 203: Clarify "T4SS modified for protein transfer".

Response 21: For clarity, we have replaced this with the specific annotation provided by the TXSScan definition to pT4SSt that refers to a specific lineage of type 4 secretion system that is capable of acting as a conjugative system and a protein transfer system.

Comment 22: - Line 210: Change "identified each in" to "identified in each of".

Response 22: Original:

“…at least 7 AMR genes were identified each in pTet subgroups 1-4.”

We have changed this to read:

“…at least 7 AMR genes were identified in each of the 4 remaining pTet subgroups.”

Comment 23: - Line 213: Change "than in" to "than".

Response 23: We have made this change.

Comment 24: - Lines 236 and 252: Clarify "protein modified T4SS".

Response 24: For clarity, we have included the annotation ID from TXSScan

Comment 25: - Lines 238-240: I recommend removing "were found among the pCC42 subgroup", which is redundant.

Response 25: We have made this change.

Comment 26: - Lines 240-242: Re-write this statement for clarity.

Response 26: We have rewritten the statement as:

"Sequence analysis of protein coding loci among plasmids from the pCC42 subgroups revealed seven loci pairs with pairwaise sequence identities >70%"

Comment 27: - Lines 256-260: Re-write these statements for clarity.

Response  27: Line 256: "tet(O) was the only AMR gene present in the pVir group, and only 3 (9.7%) of the plasmids bore the gene. "

We believe this sentence states that only ~10% of pVir plasmids carried the tet(O) gene and that the tet(O) gene was the only AMR gene present in the pVir plasmid group.

Line 257: "A 4,904bp sequence surrounding tet(O) was conserved in all pVir tet(O)+ plasmids."

We believe this sentence is clear.

Line 258: "The aligned region of this sequence had >99% sequence identity to a 7,952bp region of the reference pTet plasmid (Figure 4). "

We believe this sentence conveys that when the 4,904bp sequence in the previous sentence is aligned to the 7,952bp region of the pTet plasmid containing the same genes, there is a >99% sequence identity between the two regions.  With the aid of Figure 4, we believe this meaning is clear.

Lines 259-260: "While the aligned region contained both tet(O) and repA, repA was missing from the pVir tet(O)+ plasmids."

We see how the reader could interpret "aligned region" as "aligned sequence", which could cause confusion.  We have rewritten the sentence as "While the 7,952bp aligned region found on the pTet plasmid contained both tet(O) and repA, repA was not present on the tet(O)+ pVir plasmids."

Comment 28: - Line 277: Change "compared to" to "compared with".

Response 28: We have made this change.

Comment 29: - Lines 284 and 285: I cannot find "hybrid pT6SS/pTet plasmids"  in Table S1. Explain this.

Response 29: This is explained in the following sentence showing pT6SS plasmids containing at least 70% of the pTet reference plasmid were identified.  We believe the reader can identify these hybrid plasmids by observing the pTet reference percent identity values in column H of table S1 among plasmids from the pT6SS group (rows 169-220).   If requested, we can provide an additional supplemental table showing only the hybrid pTet/pT6SS plasmids.

Comment 30: - Lines 290 and 291: Revise this statement for clarity.

Response 30: Original lines: "In total, 16/52 of the pT6SS plasmids contained at least 70% of the reference pTet plasmid sequence and were on average 45kb larger than the 36 non-hybrid pT6SS plasmids (Figure 5)."

We have divided this into two sentences for clarity:

"In total, 16/52 of the pT6SS plasmids contained at least 70% of the reference pTet plasmid sequence.  These 16 plasmids were on average 45kb larger than the set of pT6SS plasmids that contained less than 70% of the reference pTet plasmid sequence (Figure 5)."

### Note: at this point the lines described do not correspond to the lines referenced.  We apologize if the corrections are applied to the wrong sentences following this point.

Comment 31: - Lines 300-302: This statement is confusing since "the respective ... organisms" has nothing to do with isolation source. Revise it.

Response 31: "A comparison of the four major Campylobacter plasmid groups to isolation source revealed that the respective subgroups were not found equally among Campylobacter species or host organisms (Figure 6)."

We agree with the reviewer that there is a grammatical error in the sentence, we have modified it to read:

"A comparison of the four major Campylobacter plasmid groups by isolation source revealed that the respective plasmid subgroups were not found equally among either the Campylobacter species or the host organisms from which the Campylobacter isolates were recovered (Figure 6)."

Comment 32: - Lines 304 and 305: Re-write this statement for clarity.

Response 32: Original sentence: "Further, human isolates were not represented equally among Campylobacter species between the plasmid subgroups."

We have revised this to:  "Further, the Campylobacter species composition from isolates recovered from humans differed by plasmid subgroup."

Comment 33: - Line 310: Change "four" to "four major groups".

Response 33: We have changed the sentence to read: "The pTet, pVir, pCC42 and pT6SS plasmid groups mentioned above represent the four major groups from the 30 plasmid groups that were identified."

Comment 34: - Line 312: Change "greater than" to "more than".

Response 34: The original sentence intended to communicate a value that is greater than a threshold value of five, but the reviewer’s suggestion of "more than" is not incorrect.

Comment 35: - Line 359: I recommend eliminating "(HGT)" since this acronym is not used again.

Response 35: We have made this change.

Comment 36: - Line 361: Change "encode the hallmark" to "encoded the hallmark".

Response 36: We have made this change.

Comment 37: - Lines 362 and 363: Change "contain" to "contained".

Response 37: We recognize this as a conflict between the perfect and imperfect tense.  We have revised the sentences accordingly.

Comment 38: - Line 385: Change "it also organizes" to "but it also organizes".

Response 38: We have made this change.

Comment 39: - Lines 390 and 391: Change "has been previously described" to "has previously been described".

Response 39: We agree that this phrasing is incorrect and have removed the "previously" entirely.

Comment 40: - Figure 1. I cannot find "color bars on the top". "top" should be replaced with "bottom".

Response 40: The text has been changed to reflect the current version of the figure.

Comment 41: - Figure 2. Change "to their position" to "to their positions".

Response 41: We have updated the figure text.

Comment 42: - Figure 3. I recommend eliminating "Plasmids isolated ... and 3". Change "percentage" to "ratio".

Response 42: We agree that "percentage" is incorrect and believe "proportion" is a suitable replacement.

Comment 43: - Figure 4: Change "A comparison" to "Comparison". This was mentioned in the previous review but has not been addressed.

Response 43: We have updated the figure text.

Comment 44: - Figure 5: Change "Image was" to "The image was".

Response 44: We have updated the figure text.

Comment 45: - Figure 6: Change "Bar graph representation" to "Bar graph".

Response 45: We have updated the figure text.

Round 3

Reviewer 2 Report

Comments and Suggestions for Authors

Many suggestions from the previous review have been incorporated and the current manuscript reads much better compared with the previous version. I only have minor comments as shown below.

- Lines 43 and 44: Change "whole genome sequencing" to "whole genome sequencing (WGS)".

- Lines 46 and 47: Change "schema" to "scheme" in both lines. This was mentioned in the previous review but was not properly addressed.

- Line 63: Change "Type IV" and "Type VI" to "type IV" and "type VI".

- Line 81: Change "encode for" to "encode".

- Lines 103-105: Show the number of the closed Campylobacter plasmid sequences from the NARMS program that were used in this analysis. This was mentioned in the previous review but was not addressed.

- Line 105: Change "NCBI" to "(NCBI)".

- Line 108: Change "recovered" to "extracted" or "isolated". Previously, I suggested "isolated" and that choice has nothing to do with "isolates".

- Lines 117-119: I recommend adding a sentence that defines plasmid assemblies for clarity.

- Lines 126-127: Add extra information "characterized". Characterized in what aspects?

- Line 151: Show what the adjusted Rand index indicates.

- Line 158: Change "BlastKOALA v3.0" to "with BlastKOALA v3.0".

- Line 159: I recommend changing "function" to "program".

- Line 206: Add a brief explanation for "pT4SSt".

- Line 220: Change "high number" to "large number".

- Line 226: Change "with amino acid sequence similarity to" to "similar to".

- Line 230: Change "unique variants of" to "genes encoding unique variants of".

- Line 241: Clarify and revise "protein modified T4SS". This suggestion was included in the previous review but was not properly addressed.

- Lines 245 and 246: Revise this statement for clarity.

- Line 279: Is "pVir3" a typo for "pVir.3"?

- Line 313: Change "from isolates" to "within isolates".

- Lines 318-327: No tables or figures were cited in the entire paragraph. Cite figures and/or tables.

- Line 319: Change "from the 30 plasmid groups" to "out of the 30 plasmid groups".

- Line 338: Change "closely-related" to "closely related".

- Line 339: I recommend adding legends pertaining to columns and different colors in Table S1. In fact, "TXSScan: pT4SSt, CONJScan: dCONJ_typeT" is still quite confusing.

- Line 392: Change "gene-target based" to "gene target-based".

- Table 2: Add a legend for column "% identity between variants". Does this column represent the mean percent identity?

- Figure 3: The statement "Plasmids isolated from C. coli were only represented in subgroups 1, 2 and 3." is not suitable in the figure legend. Either remove it or move it to the Results section.

Author Response

Comments 1: - Lines 43 and 44: Change "whole genome sequencing" to "whole genome sequencing (WGS)".

Response 1: We agree and have added the WGS abbreviation.

Comments 2: - Lines 46 and 47: Change "schema" to "scheme" in both lines. This was mentioned in the previous review but was not properly addressed.

Response 2: We have made this change.

Comments 3: - Line 63: Change "Type IV" and "Type VI" to "type IV" and "type VI".

Response 3: We have made this change.

Comments 4: - Line 81: Change "encode for" to "encode".

Response 4: We have made this change.

Comments 5: - Lines 103-105: Show the number of the closed Campylobacter plasmid sequences from the NARMS program that were used in this analysis. This was mentioned in the previous review but was not addressed.

Response 5: We have added this information to this section of the manuscript.

Comment 6: - Line 105: Change "NCBI" to "(NCBI)".

Response 6: We agree and have reformatted the introduction of the new abbreviation.

Comment 7: - Line 108: Change "recovered" to "extracted" or "isolated". Previously, I suggested "isolated" and that choice has nothing to do with "isolates".

Response 7: We disagree with the reviewer’s suggestion that this word requires replacement.

Comment 8: - Lines 117-119: I recommend adding a sentence that defines plasmid assemblies for clarity.

Response 8:  We have added the sentence for clarity: “The remaining contigs were processed as plasmid contigs.”

Comment 9: - Lines 126-127: Add extra information "characterized". Characterized in what aspects?

Response 9: The mob suite provides information on many characteristics of the plasmid, including GC percentage, relaxase type, replicon type, origin of transfer type, predicted mobility, mash group, predicted host range and others.  We believe the cited material provides the reader with the appropriate resources.

Comment 10: - Line 151: Show what the adjusted Rand index indicates.

Response 10: This information has been included in the Results section, lines 198-199.

Comment 11: - Line 158: Change "BlastKOALA v3.0" to "with BlastKOALA v3.0".

Response 11:  We agree and have made this change.

Comment 12:- Line 159: I recommend changing "function" to "program".

Response 12: We have made this change.

Comment 13: - Line 206: Add a brief explanation for "pT4SSt".

Response 13: “pT4SSt” is the TXSScan identifier for a Ti-plasmid type T4SS with a functional protein transfer system, and an entry has been added to the list of abbreviations.

Comment 14: - Line 220: Change "high number" to "large number".

Response 14: We agree and have made this change.

Comment 15: - Line 226: Change "with amino acid sequence similarity to" to "similar to".

Response 15: We would like to retain the original version of the statement.  The suggested modification to “pTet.2 plasmids were identified by the presence of genes encoding proteins similar to…” could imply that the function of the proteins is also similar, but we have only evaluated the similarity of the sequences.

Comment 16: - Line 230: Change "unique variants of" to "genes encoding unique variants of".

Response 16: We agree and have made this change.

Comment 17: - Line 241: Clarify and revise "protein modified T4SS". This suggestion was included in the previous review but was not properly addressed.

Response 17: we have changed this to “with a functional protein transfer system.”

Comment 18: - Lines 245 and 246: Revise this statement for clarity.

Response 18:

We have revised the sentence:

"Sequence analysis of protein coding loci among plasmids from the pCC42 subgroups revealed seven loci pairs with pairwise sequence identities >70%"

To read:

"Sequence analysis revealed seven instances where two or more protein-coding loci from pCC42 subgroup plasmids shared >70% sequence identity."

Comment 19: - Line 279: Is "pVir3" a typo for "pVir.3"?

Comment 19: “pVir3” was an earlier formatting style of “pVir.3”, we have made this change.

Comment 20: - Line 313: Change "from isolates" to "within isolates".

Comment 20: We agree and have changed this to read “within the set of isolates.”

Comment 21: - Lines 318-327: No tables or figures were cited in the entire paragraph. Cite figures and/or tables.

Response 21:  We have added citations for tables S4 and S1 in this paragraph.

Comment 22: - Line 319: Change "from the 30 plasmid groups" to "out of the 30 plasmid groups".

Response 22:  We have made this change.

Comment 23: - Line 338: Change "closely-related" to "closely related".

Response 23: We have made this change.

Comment 24: - Line 339: I recommend adding legends pertaining to columns and different colors in Table S1. In fact, "TXSScan: pT4SSt, CONJScan: dCONJ_typeT" is still quite confusing.

Response 24: We agree that the nomenclature is less than clear, however, the “pT4SSt” and “dCONJ_typeT” are the correct entries as defined by the creators of the TXSScan and CONJScan libraries.  Our inclusion of “TXSScan:” and “CONJScan:” in the column headers is intended to guide the reader to the correct MacSYFinder libraries.

We have also added a legend describing that solid colors indicate the presence/absence of the gene in the corresponding column header, color-intensity scales represent the degree of sequence similarity between the plasmid and the relationship described in the column header, and that human isolates are highlighted in the Source column.

Comment 25: - Line 392: Change "gene-target based" to "gene target-based".

Response 25: We have made this change.

Comment 26: - Table 2: Add a legend for column "% identity between variants". Does this column represent the mean percent identity?

Response 26: We agree that the column header is unclear, and it does not represent the mean percent identity.  This column represents the % identity between the amino acid sequence of the loci of the corresponding row from column 1 and column 2.  For example, Row 1 TrbE, the 91.72% value is found when translating the sequence of the locus annotated as TrbE in the pCC42.2 subgroup (corresponding to Table S2, row 15, column H) and aligning it to the translated sequence of the locus annotated as TrbE in pCC42.1 and pCC42.4 (corresponding to Table S2, row 9, column H).   These are two different loci, however they were both annotated as TrbE (or “VirB4 family type IV secretion/conjugal transfer ATPase”, or “Type IV secretion system protein virB4”, or “K20530”, depending on the reference database used).  In this way, it is the % identity of the two loci sequences and not the mean % identity between groups.  This scenario of a single name being applied to multiple sequences (or the converse situation) is a common challenge when working with virulence factor elements.  We have updated the column heading to read “% Identity between Variants 1 and 2” to communicate that the value shown is not derived from a comparison of all members of the group.

Comment 27: - Figure 3: The statement "Plasmids isolated from C. coli were only represented in subgroups 1, 2 and 3." is not suitable in the figure legend. Either remove it or move it to the Results section.

Response 27: The statement was included to assist the reader in understanding that C. coli were not present in subgroups pTet.4 and pTet.5 at rates lower than what could be observed in the bar graph representations.  However, we have removed the sentence as it did not add the intended clarity.